# 17q21.31 Microduplication Syndrome in a Patient with Autism Spectrum Disorder, Macrocephaly, and Intellectual Disability

**DOI:** 10.3390/reports6030030

**Published:** 2023-07-04

**Authors:** Federica Saia, Adriana Prato, Caterina Angela Florio, Vincenzo Paolo Cutrone, Renata Rizzo

**Affiliations:** 1Child and Adolescent Neurology and Psychiatric Section, Department of Clinical and Experimental Medicine, Catania University, 95124 Catania, Italy; adrianaprato01@gmail.com (A.P.); caterinaflo.cf@gmail.com (C.A.F.); v.p.cutrone@gmail.com (V.P.C.); rerizzo@unict.it (R.R.); 2Department of Cognitive Sciences, Psychology, Education and Cultural Studies, University of Messina, 98121 Messina, Italy

**Keywords:** 17q21.31 microduplication, genetic syndrome, psychomotor delay, intellectual disability, autism spectrum disorder

## Abstract

The chromosome 17q21.31 microduplication syndrome is a rare genetic syndrome presenting with craniofacial dysmorphisms, psychomotor delay, microcephaly, behavioral disorders, and poor social interaction. Only ten patients have been reported in the literature until today. All patients share some specific features, including psychomotor delay, behavioral disorders, and autism spectrum disorder (ASD). Here, a new case of this syndrome is reported in an 11-year-old Caucasian child who presented the classical clinical features of the 17q21.31 microduplication syndrome in association with new clinical characteristics previously unreported. The Array-Comparative Genomic Hybridization (aCGH) revealed a partial duplication of the long arm of chromosome 17. A literature review of previously studied patients with 17q21.31 microduplication syndrome is reported.

## 1. Introduction

The extensive use of Array-Comparative Genomic Hybridization (aCGH) highlighted the identification of many new genomic variants due to recurrent copy number variants (CNVs) in subjects affected by neuropsychiatric disorders like intellectual disability, behavior disturbances, and autism spectrum disorder (ASD). The chromosome 17q21.31 microduplication syndrome was first described in 2007. Kirchhoff et al. [1] reported a 10-year-old Moroccan girl with severe psychomotor delay who showed a de novo 485-kb duplication located in human chromosomal region 17q21.3.1. The other cases reported in the literature ranged in age from 6 to 18 years and shared some common clinical features like psychomotor delay, poor social interaction, and ASD. Other phenotypic observations were rather variable. The size of these duplications ranged from 485 to 763 kb. This site was already known in the scientific literature, with the corresponding microdeletion syndromes at the same chromosomal location [2]. Recently, it was reported that there are colocalized microdeletion and microduplication syndrome sites. Furthermore, a relevant number of new microduplication syndromes were described [3,4,5]. Nowadays, only ten cases of 17q21.31 microduplication syndrome have been reported in the literature [1,6,7,8,9].

The 17q21.31 microduplication syndrome is characterized by a recognizable clinical phenotype characterized by developmental delay, intellectual disability, ASD, and other clinical abnormalities, which include craniofacial dysmorphisms and hypotonia [6,7]. However, it must be admitted that the microduplication is associated with a very variable phenotype, including behavioral problems and poor social interaction; this clarifies the possibility of inheriting this mutation from a parent and not sharing the same symptoms. [7].

In this study, we report an 11-year-old male with a 17q21.31 microduplication syndrome extended for 24,858 kb from nt 43,857,094 to nt 44,105,681, inherited from his mother. In addition, the patient presented a 17q21.33 duplication, extended for 272 kb from nt 48,353,803 to nt 48,625,816. However, this last region contains interspersed high-copy repetitive sequence elements, such as short interspersed nuclear elements (SINEs), long interspersed nucleotide elements (LINEs), long terminal repeats (LTRs), and alpha satellite DNA repeats, which might promote chromosomal breakage [10,11].

Nevertheless, no rearrangement syndrome has been related to the 17q21.33 region, but three cases of 17q21.33 duplication have been reported in the literature. The first patient was reported in 1993 [12]. The 3-day-old patient presents congenital anomalies and some dysmorphic features like upslanting palpebral fissures, a large convex nasal bridge with a prominent nasal tip, and micrognathia. However, the lack of exact molecular breakpoint data precludes genotype-phenotype correlations in our case. The second one was described by Zahir et al. in 2008; the patient does not carry the same position of microduplication, and consequently, the involved mutated genes are different. [13]. In the third case, described by Kemeny in 2014, a-CGH showed a duplicated region of 0.9 Mb on chromosome 17q21.33 stretching from 48,013,468 to 48,960,310 (hg19). Our case presented some common clinical features with the patient reported by Kemeny et al., including developmental delay and mild dysmorphic elements. They also share the involvement of corresponding genes, such as EPN3, MRPL27, SPATA20, MYCBPAP, EME1, ACSF2, LRRC59, XYLT2, TMEM92, and CHAD [14].

Probably, both rearrangements reported in our proband may play a synergic role in explaining the phenotype of our case. Our proband shares some common phenotypic characteristics with previously reported patients affected by 17q21.31 microduplication syndrome, including psychomotor delay, behavioral problems, poor verbal skills, and dysmorphisms. On the other hand, he presents peculiar characteristics never described before, including macrocephaly, electroencephalogram (EEG), and brain MRI anomalies. The purpose of this study is to review the classical clinical features of the 17q21.31 microduplication syndrome [1].

## 2. Case Report

This 11-year-old Caucasian male is the firstborn of unrelated Italian parents. The mother had a history of mild developmental delay associated with mild intellectual disability. She was a housewife and attended compulsory education schools with poor performance. Moderate signs of a generalized anxiety disorder were detectable, together with autistic spectrum traits like lack of eye contact, compulsive behavior, and poor social interaction. His father was affected by obsessive-compulsive behaviors. The child was born at 41 weeks of gestation by caesarean section due to cephalopelvic disproportion. The birth weight was 3850 g (75th centile), and the height was 50 cm (50th centile). At the age of 8 months, the patient presented with her first febrile seizures, followed a few years later by other fever-induced critical episodes. In the following months, the child showed a delay in the stages of psychomotor development. Around 8 months of age, he achieved trunk control, and he started to walk independently at 24 months. In addition, the child pronounced his first word approximately at 18 months. In early childhood, the parents also described behavior problems, characterized by difficulties in social interaction and a tendency toward isolation. Considering the developmental delay, he was treated early with speech therapy and psychomotor therapy. At the age of 5 years, he was admitted for the first time to the Child and Adolescent Neurology and Psychiatry of the Medical and Experimental Department of Catania University. His weight was 27.400 g (75th centile), his height was 122 cm (50th centile), and his occipitofrontal circumference (OFC) was 54 cm (>97th centile). General conditions were good. The heart, thorax, abdomen, and internal organs were normal. He presented craniofacial dysmorphisms, consisting of distinct facial characteristics with a prominent forehead, a long face, upslanting palpebral fissures, a tubular shape of the nose, large ears, macrocephaly, synophrys, divergent strabismus, a flat and long philtrum, micro-retrognathia, bilateral knee valgus, and breech flatness. Neurologic examination revealed a lack of eye contact, language delay, and the presence of motor stereotypes (flapping hands, hopping) and tiptoe walking. During the following years, several behavioral problems became evident, including aggressivity, motor restlessness, severe speech impairment, and stereotypes. The Wechsler Intelligence Scale for Children (WISC-IV) [15] was administered and revealed the presence of mild intellectual disability (total IQ = 50). The diagnosis of ASD has been assessed using gold-standard standardized diagnostic tests, including the Autism Diagnostic Interview-Revised (ADI-R) and the Autism Diagnostic Observation Schedule (ADOS). ADI-R is an investigator-based parent or caregiver interview that gives a precise description of history and, at the same time, ongoing functioning, detecting development areas associated with autism [16]. This test suggested a high risk of an autism spectrum diagnosis. ADOS is a semi-structured, standardized evaluation of social affect that includes language, communication, social reciprocal interaction, and restricted and repetitive behaviors for individuals presumed to be affected by ASD [17]. ADOS showed difficulties in language and communication in social interaction and demonstrated restricted and repetitive behaviors. Additionally, laboratory blood testing produced normal results, including plasma and urinary amino acids, organic acids, thyroid and celiac markers, and total cholesterol. A brain MRI performed at the age of 5 years revealed an asymmetry of the posterior horns and of the lateral ventricles, with associated thinning of the posterior part of the corpus callosum. Wakeful EEG and sleep EEG showed slow and high-voltage activity.

### 2.1. Methods

This study was conducted at the Child and Adolescent Neurology and Psychiatry of the Medical and Experimental Department, Catania University. Investigations were performed as part of the routine clinical care of the patients. Prior to enrollment, written informed consent was obtained from all participants’ parents or legal guardians. Genetic testing through aCGH was performed. A DNA sample from the proband and their parents was extracted from a peripheral blood sample. aCGH was performed using the Cytosure ISCA 8 × 60K v.2 (OGT). Oxford Gene Technology (OGT, Oxford, UK) has introduced CytoSure ISCA, an aCGH designed in collaboration with the International Standard Cytogenomic Array (ISCA) Consortium. The 8 × 60k CytoSure ISCA formats provide arrays focusing on disease- and syndrome-associated genome regions, in addition to offering whole genome coverage. Using a 60-mer probe design and multiple rounds of optimization, the CytoSure ISCA ensures the detection of genetic aberrations with high signal-to-noise ratios. The genomic positions of the rearrangements refer to the public UCSC database GRCh37/hg19.

### 2.2. Results

aCGH revealed a microduplication located in 17q21.33 of roughly 272 kb. This region, not yet well characterized by scientific literature, contains high gene density and a considerable exuberance of SINEs [10]. In addition, a duplication of 24,858 kb in 17q21.31, involving four genes including *CRHR1*, *SPPL2C*, *MAPT*, and *STH* (located in intron 9 of the *MAPT* gene), responsible for the well-known 17q21.31 microduplication syndrome, was displayed. The family study revealed the presence of 17q21.31 microduplication in the mother’s proband. Based on the child’s clinical presentation and the results of aCGH, a diagnosis of 17q21.31 microduplication syndrome was made. Research in recent years has emphasized a possible connection between mutations in the PTEN gene, macrocephaly, and ASD [18]. For this reason, we have proposed to the patient’s parents more deep investigations, including the analysis of the PTEN gene, to exclude further associations that could explain the clinical phenotype. Anyway, they refused any further assessment, including PTEN gene analysis and karyotype analysis with fluorescence in situ hybridization (FISH) of the 17 chromosomes.

## 3. Discussion

The 17q21.31 microduplication syndrome represents the counterpart of the corresponding 17q21.31 microdeletion syndrome, which is characterized by developmental delay, hypotonia, facial dysmorphisms, and friendly behavior [2]. Compared to the corresponding microdeletion syndrome, 17q21.31 microduplications have been seen less frequently. Kirchhoff et al. [1] described the frequency among live births of 1/55,000 and 1/327,000 for microdeletions and microduplications, respectively, which suggests a ratio of 6:1, lower than expected for non-allelic homologous recombination (NAHR). Ten cases of 17q21.31 microduplication syndrome have been previously described in the literature [1,6,7,8,9]. Our proband showed psychomotor delay, ASD, mild intellectual disability, and both craniofacial and body dysmorphisms. In addition, his mother presented a psychiatric profile with an unspecified intellectual disability, ASD traits, and anxiety. Despite other cases, our proband presented with macrocephaly associated with MRI and EEG abnormalities, characteristics never reported in the literature (Table 1). The duplicated sequence of our proband’s DNA contains four genes: *CRHR1*, *SPPL2C*, *MAPT*, and *STH*. The *MAPT* (microtubule-associated protein TAU) gene encodes proteins that stabilize the microtubules and, in some mutations, could be associated with neuropsychiatric disorders. Moreover, the *MAPT* gene is implicated in the mediation of autism-like behaviors by impairing myelination in oligodendrocytes and synaptic function in neurons [19]. On the other hand, some studies have described MAPT duplication in neurodegenerative disorders. Mutations in the microtubule-associated protein tau (MAPT) encompass multiple neurodegenerative disorders, but the pathophysiological mechanisms remain unclear. A novel variant in MAPT resulting in an alanine-to-threonine substitution at position 152 (A152T tau) has recently been described as a significant risk factor for both frontotemporal lobar degeneration and Alzheimer’s disease [20]. MAPT duplication is supposed to be the cause of early-onset idiopathic neurodegenerative dementia or early-onset idiopathic atypical extrapyramidal syndromes [21]. Chen et al. in 2019 [22] demonstrated the involvement of MAPT gene duplication in progressive supranuclear palsy. The potential link between MAPT gene duplication and early-onset uncommon dementia with tau accumulation should be further investigated and may correlate with neuroimaging findings. *SPPL2C*, *MAPT*, and *STH* are also involved in the development of Frontotemporal dementia (FTD), as described by Ferrari et al. in 2017 [23]. The *CRHR1* (corticotropin-releasing hormone receptor 1) gene encodes a G-protein-coupled receptor that regulates the hypothalamic-pituitary-adrenal pathway. A duplication of this gene could contribute to behavioral problems and poor social interaction [24]. The *CRHR-1* gene contributes to the expression of ASD symptoms; furthermore, CRH levels are increased in the serum of children with ASD as part of a ‘brain-stress-immunity’ connection [25]. The *SPPL2C* (Signal Peptide Peptidase Like 2C) gene is a protein-coding gene that enables protein homodimerization activity and is involved in membrane protein proteolysis. This gene is associated with chromosome 17q21.31 microduplication syndrome [9]. The *STH* gene is a polymorphic gene nested within an intron of the *MAPT* gene and encodes the protein Saitohin, which has been found to be susceptible to multiple degenerative diseases and neuropsychiatric disorders [26].

Although we did not investigate TAD boundaries and the effect of their disruption, they can play a role in the different clinical phenotypes and heterogenous manifestations of this microduplication. TADs are genomic regions in which genes share regulatory elements that are functionally separated by nearby domains. It is well established that structural abnormalities spanning coding genes can have pathogenic consequences due to gene dosage effects or alter the expression of nearby genes. However, it remains unclear how alterations in TAD structure can lead to disease etiology [27].

In 2017, Arbogast et al. generated mouse models to describe the pathophysiology of these microduplications [28]. They studied the deletion and duplication of the syntenic region to deeply investigate the syndromes. They found a single phenotypic similarity between patients carrying the 17q21.31 microduplication and Dup/+ mice, which is microcephaly that has been reported in a considerable number of human individuals with this microduplication. A lot of SNPs associated with risk for Alzheimer’s disease (AD) were identified near MAPT and KANSL1 in humans, and they appeared to be correlated with an overexpression of both genes in different brain regions [23]. This observation was further supported by the description of a familial form of late-onset AD due to the microduplication of the 17q21.31 region [29].

To the best of our knowledge, this case report increases the number of reported cases in the literature to 12 patients with 17q21.31 microduplication syndrome (Table 1). Due to the low number of cases described, it is difficult to draw a clear picture of the characteristics of this microduplication. Conversely, a consistent number of people with microduplication who do not present the typical clinical features discover the microduplication after the diagnosis in the proband. Our patient has similar characteristics to patient 3 reported by Grisart et al. [7]. Both phenotypes present common clinical features, including tiptoe walking, facial dysmorphism, motor delay, and an ASD diagnosis. Compared with other reported cases, the IQ level of our child is the lowest (IQ = 50). Moreover, our case showed behavioral problems like aggressivity, motor restlessness, and stereotypes, in line with other reported cases, except for the proband and his mother described by Natacci et al. [6]. Our proband presented many facial dysmorphisms, like micrognathia, in common with the patients described by Kirchhoff [1] and Kitsiou-Tzeli [8], and synophrys like patients 1 and 2 described by Grisart et al. [7]. The main differences between these cases and our proband appeared to be MRI and EEG results, the presence of macrocephaly, and IQ level. Concerning the few cases reported in the literature, no one showed EEG or MRI abnormalities. Instead, our proband’s MRI showed asymmetry of the posterior horns and of the lateral ventricles and a light thinning of the posterior part of the corpus callosum. In addition, wakeful EEG and sleep EEG showed slow and high-voltage activity. Another important difference is the size of the duplication. Previous cases reported a size between 485 and 763 kb, while our case holds a duplication size of 24,858 kb. Nevertheless, our case involves a second rearrangement on the same chromosome, a 17q21.33 duplication. This CNV, until today, was not associated with a specific syndrome; however, three cases were just reported in the literature. Among these three cases, only Kemeny’s patient [14] shared some common features with our patient, including developmental delay and mild dysmorphic elements. This region is not well characterized by scientific literature but comprises high gene density (including TMEM92, XYLT2, LRRC59, MRPL27, EMEL, ACSF2, CHAD, RSAD, MYCBPAP, EPN3, and SPATA20), dosage-sensitive genes, an excess of segmental duplications, and a relative abundance of SINEs, which predispose this chromosome to genomic rearrangements and could be responsible for clinically relevant phenotypes [10]. We suggest TMEM 92 and ACSF2 (involved in brain lipidic metabolism) as possible candidate genes for central nervous system impairment [30]. CHAD could be a possible candidate gene for musculoskeletal disorders. CHAD (chondroadherin) is a non-collagenous extracellular matrix protein important in the regulation of chondrocyte organization within cartilage [31]. The duplicated region reported in our patient is smaller than the duplicate region of 0.9 Mb described by Kemeny et al. [14]. The COL1A gene is in the non-overlapping regions; this could explain the principle phenotypic differences between our patient and Kemeny’s patient, the severe kyphoscoliosis [14]. Both rearrangements in our proband could play a role together to explain the phenotype.

## 4. Conclusions

In conclusion, it is possible that cases of 17q21.31 microduplication syndrome have been under-ascertained due to their milder and various phenotypes and later onset. Undoubtedly, further studies are essential to better delineate the genetic background of patients with 17q21.31 microduplication, clarify the complex genetic structure of these syndromes, better define the outcome, predict early pathological signals, and identify new possible therapeutic targets. Nevertheless, more exploration is required to understand the molecular mechanisms involved. More cases are also needed to better understand the 17q21.33 duplication phenotype and how it could influence other genomic imbalances.

## Figures and Tables

**Table 1 reports-06-00030-t001:** Molecular and clinical characteristics of 17q21.31 microduplication syndrome in previously reported patients compared to our patients.

	Our Case	Mother of the Case	Kirchhoff et al., 2007 [1]	Grisart et al., 2009 [7]				Kitiou-Tzeli et al., 2012 [8]	Mc Cormack et al., 2014 [9]	Natacci et al., 2015 [6]		
				Case 1	Case 2	Case 3	Case 4			Case 1	Case 2 (Mother)	Case 3 (Maternal Uncle)
Chromosomal Microduplication region	GRCh37/hg19 (438,570,944–4,105,681) × 3	ND	GRCh37/hg19 (431,675,408–44,159,862)	ND	ND	ND	ND	ND	GRCh37/hg19 (43,645,879–44,292,742)	ND	ND	ND
Origin of the duplication	Inherited	ND	Inherited paternal	De novo	De novo	De novo	ND	Inherited paternal	ND	Inherited maternal	Inherited	Inherited
Sex	M	F	F	F	F	M	M	F	F	F	F	F
Age at report	11 years old	ND	ND	6 years old	6 years old	4 years old	ND	18 years old	2 years and 7 months old	4 years old	ND	ND
Ethnicity/Nationality	Italian	Italian	Moroccan	ND	ND	ND	ND	Greek	Iraqi-Afghan	Italian	Italian	Italian
Gestational age	At term	ND	42 weeks	At term	At term	At term	At term	At term	At term	At term	ND	ND
Birth Growth Parameters: weight (g), length (cm), and OFC (cm)	3850 g (75th centile); 50 cm (50th centile)	ND	3070 g (10th–25th centile); 50 cm (25th centile); 34 cm (25th–50th centile)	3570 g (50th centile)	3500 g (50th centile); 53 cm (90th centile);34.2 cm (25th–50th centile)	2890 g (−2SD); 50 cm (50th centile); 34 cm (10th centile)	5100 g (+3SD); 58 cm (+3SD)	2500 g (10th centile); 48 cm (25th centile); 33 cm (10th–25th centile)	2490 g (3rd–10th centile)	3530 g (90th centile); 48 cm (<10th centile); 33 cm (<10th centile)	ND	ND
Developmental Delay/ID	+	+	+	+	+	+	+	+	+	+	+	+
Autism spectrum disorder	+	ND	-	-	-	+	+	-	-	-	-	-
Hypotonia	-	ND	+	-	+	-	+	-	-	-	-	-
Behavioral disorders	+	+	ND	-	+	-	+	+	-	-	-	+
Microcephaly	-	ND	+	+	-	-	-	-	+	+	+	-
Macrocephaly	+	ND	-	-	-	-	-	-	-	-	-	-
Brain anomalies	Asymmetry of the posterior horns and of the lateral ventricles and light thinning of the posterior part of the corpus callosum	ND	ND	-	ND	-	ND	-	ND	ND	ND	ND
Dysmorphism	Prominent forehead, long face, upslanting palpebral fissures, tubular shape of the nose, large ears, synophrys, divergent strabismus, flat and long philtrum, and micro-retroignathia	ND	Ears with unfolded helixes, a short nose with a prominent nasal tip and columella, a smooth philtrum, a small mouth with a high vaulted and narrow palate, dental malposition, micrognathia, short and broad thumbs, terminal broadening of fingers, a long first toe, and hirsutism on the back	Synophrys, dysplastic ears, puffy eyelids, short philtrum, thin upper lip, micrognathia, clinodactyly of the fifth finger, single palmar crease on the right hand, partial bilateral syndactyly, and slight hirsutism on the back	Epicanthus, large posteriorly rotated ears, short upturned nose, short philtrum, flat midface, high arched palate, dental malposition, low posterior hairline, thick hair, and tapering fingers	Brachycephaly, open square facies with pointed chin, and a congenital hypopigmented 3 × 3 cm mark on flank	Mild malar hypoplasia, simple and elongated ears, mild pectus excavatum, clinodactyly 5th, 2–3 syndactyly, sandal gap, and a slender body habitus	Short nose, prominent nasal tip and columella, philtrum smooth, small mouth, mild micrognathia, global hirsutism, hemangioma of the temporal region, and strabismus	Almond eyes and small hands	Palpebral fissures down, prominent nasal tip, flat midface, small mouth, dental malposition, and tapering hands		Palpebral fissures down, puffy eyelids, short nose, flat midface
Skeletal/limb anomalies	Bilateral knee valgus and breech flatness	-	-	ND	-	-	-	-	ND	ND	ND	ND
Growth (obesity/FTT)	Normal	Normal	FTT	Mild truncal obesity	Normal	Normal	Normal	Obesity	Normal	Obesity	Obesity	Obesity
Other	EEG: high voltage during the sleep and awake phases		Atopic dermatitis and sleeping problems				Hypogonadism	Ataxic gait and VSD				

FTT: failure to thrive; ND: not determined; SD: standard deviation; “+”: trait present in the patient; “-”: trait not observed in the patient; EEG: electroencephalogram; VSD: ventricular septal defect.

## Data Availability

The data presented in this study are available on request from the corresponding author.

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
