# Peer review of "17q21.31 Microduplication Syndrome in a Patient with Autism Spectrum Disorder, Macrocephaly, and Intellectual Disability"

_reports, 2023, doi:10.3390/reports6030030_

Round 1

Reviewer 1 Report

1 spell out all abbreviations

2. minor language corrections are necessary

3. Please extend the references section and introduction and Discussion section as well. 

Reconsider after major

Author Response

Independent Review Report, Reviewer 1

Comments and Suggestions for Authors:

1. spell out all abbreviations.

2. minor language corrections are necessary.

3. Please extend the references section and introduction and Discussion section as well.

We thank the reviewer for all useful comments and suggestions. We are extremely grateful for your time spent reviewing our article. First, we added the minor language correction requested. We also spelled out all the abbreviations used in the manuscript. Finally, we followed your suggestion and expanded the introduction and discussion section, including more references regarding the previously reported literature cases of 17q21.31 microduplication syndrome.

Reviewer 2 Report

Report on 9th case with microduplication 17q21.31 is overall fine

Still it seems to be at least the 11th case acc. to https://www.orpha.net/consor/cgi-bin/OC_Exp.php?lng=EN&Expert=217340 - see leavelet of Unique from 2013

and acc. to pubmed 17q21.31 microduplication syndrome - Search Results - PubMed (nih.gov)  there should be more cases - even such being possible ignored before by Kirchhoff et al. - check and include

Als the mouse model for the disorder needs to be mentioned - PMID: 28704368

things to be fixed further:

- check whlole paper for writing errors and uniform nomenclature - e.g. there are several ways how array-CGH is termed, intead of "11-years-old child" it must read as "11-year-old child", line 31 delete "1" after Kirchhoff et al.1 reported, line 32 there is two times 'known', citation is funny like e.g. in line 36 as ".(2); (3), (4),", why is "Obsessive Compulsive Behaviour" and "Electroencephalogram" written with capital letters? Include explanation for abbreviations at first use - e.g. lacking for ASD or FISH. 

- lines 41-48 - what buildt are you referring to? hg19, hg37, hg38? 

- line 60: gramm is abbreviated as g 

- avoid things which could identify the patient like Italian parents or places where the patient was treated or studied

- gene names have to be written in italics

- Table 1 is obviously a screenshot with red underlinings from a word document - this cannot be submitted like that. Also there are writing errors like '2-3syndactyly' instead of '2-3 syndactyly' or comma after "hairlaine"

- ethical statement can be removed from text part as it is mention in declarations in the end of paper 

see above

Author Response

We thank the reviewer for all useful comments and suggestions. We are extremely grateful for your time spent reviewing our article. The following are our answers point by point to your issues. First, we included the other reported literature cases of 17q21.31 microduplication syndrome. Following your advice, we also mentioned the paper written by Arbogast ed al. regarding the mouse models of 17q21.31 microdeletion and microduplication syndromes. Furthermore, we added the minor language correction requested and we removed the ethical statement from the main text. Finally, we fixed table 1 in the correct format, as you suggested.

Reviewer 3 Report

Please see my comments attached.

Author Response

We thank the reviewer for all useful comments and suggestions. We are extremely grateful for your time spent reviewing our article. The following are our answers point by point to your issues. First, we added the minor language correction requested in the abstract and introduction section, as you recommended. Second, we mentioned the percentile for weight and length in the case description. We spelled out all the abbreviations used in the manuscript, including in the table 1. Considering that the 17q21.31 microduplication was also detected in the mother of the proband, we followed your suggestion including her clinical features to the table 1.

Furthermore, we agree with the reviewer regarding the importance of MRI images

Unfortunately, we cannot provide a picture of the patient. Moreover, we better clarified why we have proposed to patient’s parents the analysis of PTEN gene. Research in recent years has emphasized a possible connection between mutations in PTEN, macrocephaly and ASD. For this reason, we suggested deep investigations, including the analysis of the PTEN gene, to exclude further associations that could explain the clinical phenotype. Anyway, they refused any further assessment including PTEN gene analysis and karyotype analysis with fluorescence in situ hybridization (FISH) of 17 chromosome.

As you recommended, we edited and expanded the discussion section, including the other reported literature cases of 17q21.31 microduplication syndrome.Please see the attachment. Finally, we modified and fixed table 1 in the correct format, as you suggested.

Our Case

Mother of the Case

Kirchhoff et al, 2007

Grisart et al, 2009

Kitiou-Tzeli et al, 2012

Mc Cormack et al, 2014

Natacci et al, 2015

Case 1

Case 2

Case 3

Case 4

Case 1

Case 2 (mother)

Case 3 (maternal uncle)

Chromosomal Microduplication Size

248,58kb

     ND

485kb

From 585kb to 763 kb

From 585kb to 763 kb

From 585kb to 763 kb

ND

694.6 kb

647kb

689kb

ND

ND

Origin of the duplication

Inherited

ND

Inherited paternal

De novo

De novo

De novo

ND

De novo

ND

Inherited maternal

Inherited

Inherited

Sex

M

F

F

F

F

 M

M

F

F

F

F

F

Age at report

11 years old

ND

ND

6 years old

6 years old

4 years old

ND

18 years old

2 years and 7 months old

4 years old

ND

ND

Ethnicity/Nationality

Italian

Italian

Moroccan

ND

ND

ND

ND

Greek

Iraqui-Afghan

Italian

Italian

Italian

Gestational age

At term

ND

42weeks

 At term

At term

At term

At term

At term

At term

At term

ND

ND

Birth Growth Parameters: weight(g), length(cm), OFC (cm)

3850g (75th centile)

50 cm (50th centile)

ND

3070 g (10th -25th centile)   

50 cm (25th centile)

34 cm (25th -50th centile)

3570 g (50th centile)

3500 g (50th centile)

53cm (90th centile)                         34.2 cm(25th -50th centile)

2890g (-2SD)

50cm (50th centile) 34cm (10th centile)

5100 g(+3SD)

58 cm(+3SD)

2500g (10th centile)

48 cm (25th centile)

33 cm (10th -25th centile)

2490 g (3rd -10th centile)

3530 g (90th centile)

48 cm (<10th centile)

 33 cm(<10th centile)

ND

ND

Developmental Delay/ID

+

+

+

+

+

+

+

+

+

+

+

+

Autism spectrum disorder

+

ND

-

-

-

+

+

-

-

-

-

-

Hypotonia

-

ND

+

-

+

-

+

-

-

-

-

-

Behavioral disorders

+

+

ND

-

+

-

+

+

-

-

-

+

Microcephaly

-

ND

+

+

-

-

-

-

+

+

+

-

Macrocephaly

+

ND

-

-

-

-

-

-

-

-

-

-

Brain anomalies

Asymmetry of the posterior horns and of the lateral ventricles and light thinning of the posterior part of corpus callosum

ND

ND

-

ND

-         

ND

-

ND

ND

ND

ND

Dysmorphism

Prominent forehead long face, up slanting palpebral fissures, tubular shape of the nose, large ears, synophrys,divergent strabismus,flat and long philtrum,micro-retroignathia

ND

ears with unfolded helixes, short nose with prominent nasal tip and columella, smooth philtrum, small mouth with a high vaulted and narrow palate, dental malposition, micrognathia, short and broad thumbs, terminal broadening of fingers, long first toe,

hirsutism on the back

synophrys, dysplastic ears, puffy eyelids, short philtrum, thin upper lip, micrognathia, clinodactyly of 5th finger,

single palmar crease on right hand,

partial bilateral syndactyly, slight hirsutism on the back

epicanthus, large posterior rotated ears, short upturned nose, short philtrum, flat midface, high arched palate,

dental malposition, low posterior hairline, thick hair, tapering fingers

Brachycephaly, open square facies with pointed chin, congenital hypopigmented 3x3cm mark on flank

Mild malar hypoplasia, simple and elongated ears, mild pectus excavatum, clinodactyly 5th, 2-3 syndactyly and sandal gap. Slender body habitus

Short nose, prominent nasal tip and columella, philtrum smooth, small mouth, mild micrognathia, global hirsutism, hemangioma of temporal region, strabismus

Almond eyes, small hands

Palpebral fissures down, prominent nasal tip, flat midface, small mouth, dental malposition, tapering hands

Palpebral fissures down, puffy eyelids, short nose, flat midface

Skeletal/limbs anomalies

Bilateral knee valgus and breech flatness

-

-

ND

-

-

-

-

ND

ND

ND

ND

Growth (obesity/FTT)

normal

normal

FTT

mild truncal obesity

normal

normal

Normal

obesity

normal

obesity

obesity

obesity

other

EEG: high voltage during the sleep and awake phases

Atopic dermatitis, sleeping problems

Hypogonadism

Ataxic gait, VSD

Round 2

Reviewer 1 Report

Well adressed suggestions 

Author Response

We are extremely grateful for your time spent reviewing our article

Reviewer 2 Report

Authors worked in the paper

still to fix:

- authors need to use a uniform nomenclature for aCGH approach - e.g. only using 'array-comparative genomic hybridization (aCGH) instead of many different variants yet used in the text

- authors need to do a propper literature research - they still miss  cases from PMID: 25106685 (this report includes CHAD gene duplication), https://www.sciencedirect.com/science/article/pii/S1769721214001505 (includes COL1A1, SGCA, PPP1R9B and CHAD), PMID: 19449402 (includes CHAD as well); more reports are listed by UNIQUE under https://www.rarechromo.org/media/information/Chromosome%2017/17q21.31%20duplications%20FTNW.pdf

- as reports of authors also include MAPT gene duplications also such cases need to be included - and /or at least the signs and symptoms of MAPT duplication, to understand what is a result of MAPT gene and possible other ones. Also TAD-analyses should be done. It is nowadays well known that TAD disruption from a farer region can influence teh outcome and here it is not excluded yet that the syndrome  reported may just be a variant of the 17q21.31 duplication syndrome.  

- Table 1 - give the localization of the duplicated regions rather than their size. Given them all in the same buildt - e.g. GRCh37/hg19

Author Response

Independent Review Report, Reviewer 2

-authors need to use a uniform nomenclature for aCGH approach - e.g. only using 'array-comparative genomic hybridization (aCGH) instead of many different variants yet used in the text

- authors need to do a propper literature research - they still miss  cases from PMID: 25106685 (this report includes CHAD gene duplication), https://www.sciencedirect.com/science/article/pii/S1769721214001505 (includes COL1A1, SGCA, PPP1R9B and CHAD), PMID: 19449402 (includes CHAD as well); more reports are listed by UNIQUE under https://www.rarechromo.org/media/information/Chromosome%2017/17q21.31%20duplications%20FTNW.pdf

- as reports of authors also include MAPT gene duplications also such cases need to be included - and /or at least the signs and symptoms of MAPT duplication, to understand what is a result of MAPT gene and possible other ones. Also TAD-analyses should be done. It is nowadays well known that TAD disruption from a farer region can influence teh outcome and here it is not excluded yet that the syndrome  reported may just be a variant of the 17q21.31 duplication syndrome.  

- Table 1 - give the localization of the duplicated regions rather than their size. Given them all in the same buildt - e.g. GRCh37/hg19

We thank the reviewer for all useful comments and suggestions. We are extremely grateful for your time spent reviewing our article. The following are our answers point by point to your issues.

First, we used a uniform nomenclature for a CGH (only 'array-comparative genomic hybridization (aCGH)

Then we included more literature research as you suggest.

Also, we added MAPT gene duplication and the influence of TAD analyses.

Finally, we added all the required field in table1, except for the case of Grisart et al, where only a schematic representation of genomic duplication was shown (see the attachment), so we had troubles to deduce the exact duplicated region.

Reviewer 3 Report

The manuscript looks much better. Line 64: Add Italian descent. Line 79: What does is mean" At 5 years old, he has been taken in charge to Child and Adolescent...etc"?

Add the refereneces for all reported case in line 146

There is mispellin9 in line 186: "idented=identified"? and line 196" "synophrisys= synophrys"

English revision is still needed including punctuation and grammar.

Author Response

Independent Review Report, Reviewer 3

The manuscript looks much better. Line 64: Add Italian descent. Line 79: What does is mean" At 5 years old, he has been taken in charge to Child and Adolescent...etc"?

Add the refereneces for all reported case in line 146

There is mispellin9 in line 186: "idented=identified"? and line 196" "synophrisys= synophrys"

We thank the reviewer for all useful comments and suggestions. We are extremely grateful for your time spent reviewing our article.We added Italian descent. Then in line 79, we wrote when the boy came to our attention for the first time. Additionally, we have changed the misspellings as you suggest .Finally we added the references requested and we made English revision.

Round 3

Reviewer 2 Report

ok thanks

Author Response

We thank the reviewer for all useful comments and suggestions. We are extremely grateful for your time spent reviewing our article.